# Pre-Clinical Models for CAR T-Cell Therapy for Glioma

**DOI:** 10.3390/cells13171480

**Published:** 2024-09-04

**Authors:** Gust Vandecandelaere, Rishab Ramapriyan, Matthew Gaffey, Leland Geoffrey Richardson, Samuel Jeffrey Steuart, Masih Tazhibi, Adrian Kalaw, Eric P. Grewal, Jing Sun, William T. Curry, Bryan D. Choi

**Affiliations:** 1Brain Tumor Immunotherapy Lab, Department of Neurosurgery, Massachusetts General Hospital, Harvard Medical School, Boston, MA 02114, USA; gvandecandelaere@mgh.harvard.edu (G.V.);; 2Faculty of Medicine, KU Leuven, 3000 Leuven, Belgium

**Keywords:** CAR T-cell therapy, pre-clinical models, immunotherapy, solid tumors, glioma

## Abstract

Immunotherapy represents a transformative shift in cancer treatment. Among myriad immune-based approaches, chimeric antigen receptor (CAR) T-cell therapy has shown promising results in treating hematological malignancies. Despite aggressive treatment options, the prognosis for patients with malignant brain tumors remains poor. Research leveraging CAR T-cell therapy for brain tumors has surged in recent years. Pre-clinical models are crucial in evaluating the safety and efficacy of these therapies before they advance to clinical trials. However, current models recapitulate the human tumor environment to varying degrees. Novel in vitro and in vivo techniques offer the opportunity to validate CAR T-cell therapies but also have limitations. By evaluating the strengths and weaknesses of various pre-clinical glioma models, this review aims to provide a roadmap for the development and pre-clinical testing of CAR T-cell therapies for brain tumors.

## 1. Introduction

### 1.1. Background

The development of chimeric antigen receptor (CAR) T-cell therapy for cancer has progressed dramatically since its first generation in 1993 [1,2,3,4]. CAR T-cell therapy is based upon the expression of CARs on T cells, which are engineered to target specific target antigens and trigger T-cell activation pathways upon binding. By directly recognizing target antigens, CAR T cells do not require traditional mechanisms of antigen presentation; therefore, they act in an HLA-independent manner. This recognition is mediated by the extracellular antigen-binding domain of the CAR, which activates the intracellular signaling domains in the T cell. CAR T-cell therapy has been successful in treating hematological cancers but has had limited efficacy against solid tumors [5]. Indeed, all currently FDA-approved CAR T-cell therapies are treatments for forms of leukemia, lymphoma and multiple myeloma [6]. Solid tumors are difficult malignancies to treat due to tumor heterogeneity, an unfavorable tumor microenvironment (TME) and concomitant toxicities [7,8,9]. Recently, the first immune cell therapy for solid tumors was approved by the FDA. Amtagvi (lifileucel), developed by Iovance Biotherapeutics, is a tumor-infiltrating lymphocyte (TIL) cancer therapy that has shown success in treating melanoma [10].

Malignant brain tumors encompass a wide range of subtypes and classifications [11]. For most of these, available treatments are non-curative and have many unwanted side-effects. CAR T-cell therapies have recently shown promise in treating brain tumors [12,13]. Despite this, they face significant barriers to further success—such as tumor heterogeneity, a hostile TME and impaired delivery across the blood–brain barrier (BBB)—that may be addressed in the pre-clinical research stage. Despite these challenges, research in this field has made several advances in recent years, including combination therapies, armored CAR T cells and multi-targeting constructs. These platforms require robust pre-clinical models to test efficacy and recapitulate the most salient aspects of their target malignancies (Figure 1).

Pre-clinical models allow for the screening and evaluation of candidate therapies, particularly for toxicity and efficacy prior to clinical translation. The bulk of pre-clinical work is typically conducted in murine models, which have several strengths and limitations depending on the specific approach. Common models, such as genetically engineered murine models (GEMMs), transgenic mice, and canine and primate models, offer new perspectives. Careful selection and combination of both in vitro and in vivo testing must be considered. This should be based on factors such as the recapitulation of the human TME and the ability to optimally account for associated toxicities in humans. This review describes the different types of currently available pre-clinical models for gliomas, including their qualities and limitations, that can be leveraged to develop new CAR T-cell therapies for tumors in the brain. Similar experimental considerations could be applied to models for TIL therapy.

### 1.2. CAR T-Cell Therapy for Glioma

Recent advances have shown great promise for CAR T-cell therapy for solid brain tumors [14]. CAR T cells have been shown to cross the BBB and, in some instances, to do so more effectively than other therapeutic strategies, such as monoclonal antibodies or vaccines [15,16]. Another benefit of CAR T-cell therapy is the ability to directly kill tumor cells without the need for an endogenous immune response. Currently, many CAR T cells are in pre-clinical development for solid brain tumors, aiming to successfully translate into clinical trials [17,18].

The most prevalent and lethal type of malignant brain tumor is glioblastoma (GBM). It is classified as a grade 4 tumor according to the World Health Organization (WHO) grading system. The median survival for patients diagnosed with glioblastoma is close to 15 months, highlighting the urgency of developing novel therapies [19]. GBM is the most aggressive type of glioma, which is a broader category of brain tumors. Gliomas are thought to arise from glial cells, which are supportive cells that surround and provide critical support for neurons in the brain, spine or related progenitor cells. Not all gliomas are considered malignant, as some grow slowly and are not considered to be cancerous. Anaplastic gliomas, diffuse intrinsic pontine gliomas (DIPGs) and glioblastomas are some examples of malignant gliomas.

There are several limitations in the treatment of glioma with CAR T cells. First, the identification of adequate tumor-associated antigens (TAAs) has been difficult, as GBMs are heterogeneous and promote antigen escape. Moreover, such targets are often not exclusively found in tumor tissue. This carries the theoretical risk of on-target/off-tumor toxicity, which can be life-threatening [20,21]. Secondly, the hostile TME for CAR T cells represents a significant barrier. A number of tumor-associated changes, including hypoxia, immune infiltration, the Warburg effect and hostile metabolism, combine to promote tumor survival while also inhibiting the efficacy of T cell-based therapies [17].

Prior pre-clinical and clinical studies of CAR T cells that target solid brain tumors have demonstrated the potential for therapeutic benefit. Promising results have been achieved in treating diffuse intrinsic pontine glioma with CAR T cells targeting GD2 [22]. Another target, B7-H3, was shown to have encouraging outcomes in a clinical trial [23]. Another study leveraging CAR T-cell therapy for patients with high-grade glioma targeted EGFRvIII engineered to secrete T cell-engaging antibodies that bind to wild-type EGFR [24,25,26,27]. Tumors regressed in all three patients treated with this therapy. Previously, this unique approach had been tested using several different xenograft models and pre-clinical assays to assess safety, efficacy and toxicity [28].

While CAR T-cell therapy can be highly effective, it can result in life-threatening toxicities. This is of great importance, as pre-clinical models often do not effectively predict these. Toxicities can present with many symptoms. Cytokine Release Syndrome (CRS) and Immune effector Cell-Associated Neurotoxicity Syndrome (ICANS) are most commonly found in patients [29]. Syngeneic murine models have not accurately recapitulated the toxicities observed in patients treated with CAR T-cell therapy, which is reported in 37–93% of patients across various studies [30]. Among pre-clinical publications of CAR T-cell therapy for glioma, very few report on toxicity, despite its observation in the clinical setting [31].

## 2. Overview of CAR T-Cell Generation

The correct generation of CAR T cells is crucial to effectively implement these pre-clinical models for glioma. Here, we discuss the choice of good candidates as TAAs for CAR T-cell therapy, as well as how researchers can create or obtain their own CAR T-cell therapies.

### 2.1. Choosing CAR T Cells

CAR T cells should be engineered to target antigens that are overexpressed on glioma cells specifically. Common targets include EGFRvIII, IL13Rα2, HER2 and GD2. As previously discussed, it is challenging to find adequate glioma TAAs for CAR T-cell therapy. In glioma, TAAs are often not tumor-specific but can also be found in small amounts in healthy tissue. Consequently, CAR T-cell therapies targeting these antigens carry the theoretical risk of harm to non-tumor tissues. Researchers are developing new methods to address these challenges and effectively eliminate glioma cells, such as CAR T cells secreting BiTEs or dual TAA targeting [28,32].

### 2.2. Generating CAR T Cells

Detailed protocols are available in the literature that comprehensively describe various processes of generating customized CAR T cells [33,34]. This can be a rigorous process involving multiple precise steps, and often, collaboration with experts in cell therapy and gene engineering is necessary in order to successfully accomplish these aims. First, an antigen recognition domain should be designed that is specific to the chosen antigen and translated in tandem with a spacer/linker, as well as transmembrane and signaling domains [35]. Several iterations of CARs have been designed, and new generations are actively being engineered [36]. The CAR gene must be subsequently incorporated into a gene delivery platform. The typical delivery strategy is viral transduction due to its high transduction efficiency. Lentiviral transduction is most commonly used for CAR T cells [37].

However, non-viral alternatives are actively being explored to mitigate the higher immunogenicity and manufacturing cost associated with viral vectors [38]. These include the Sleeping Beauty (SB) transposon system, designer nucleases, CRISPR/Cas9 systems, electroporation and nanoparticles [39]. The development and delivery of CAR T-cell therapies could become more accessible and efficient through the use of these methods [40]. Using CRISPR/Cas9, researchers demonstrated high levels of safety and efficiency for anti-CD19 PD1 CAR T cells in pre-clinical models and a phase I clinical trial [41]. SB-engineered CAR T cells were also successfully employed in a phase I/II clinical trial without severe toxicities [42].

Depending on the selected model, T cells should also be derived from the appropriate origin. For example, when leveraging a syngeneic mouse model, where both the immune system and target tumor cells are murine, CAR T cells should be made with murine T cells. However, when using a xenograft mouse model with human target tumor cells, human CAR T cells should be used.

Another possible source of CAR T cells for researchers is transgenic mouse lines developed by the Jackson Laboratory (JAX). These mice are genetically modified so that T cells express specific human or murine proteins, facilitating pre-clinical models. However, as CAR T-cell targets for glioma are delicate and new strategies are continuously being explored, researchers generally generate their own CAR T cells [43]. Researchers have developed unique mouse strains, for example, those from which all T cells continuously express EGFRvIII and Her2 CAR [44,45]. Making murine CAR T cells can be a laborious process and may lead to batches of differing quality. This strategy could serve as a useful source of reliable murine CAR T cells that can be used in syngeneic mouse models.

## 3. Overview of Glioma Tumor Models for CAR T-Cell Therapy

In this section, we discuss several glioma models that have been used for the testing of CAR T-cell therapy.

### 3.1. Cell Lines

#### 3.1.1. Murine Glioma Models

Murine cell lines for malignant glioma are used in syngeneic mouse models. Because these models have an intact immune system, they require cells to be injected into mice of an analogous genetic background. This offers a significant advantage in addressing questions to which the endogenous immune environment may be relevant. Different strategies exist to generate glioma cell lines, including chemical induction, viral induction, the use of Sleeping Beauty transposons and the isolation of spontaneously arising tumors.

Chemically induced glioma tumor models include GL261 and CT-2A. These tumor lines were created using carcinogenic chemical compounds. The first brain tumor was induced in a mouse via intracranial delivery of 20-methylcholanthrene in the C3H mouse strain in 1939 [46]. These types of models are especially vulnerable to genetic drift and high mutational burdens [47]. Chemically induced models also frequently lack mismatch repair (MMR) genes, which make cells more robust against alkylating agents such as temozolomide. Virally induced tumor cell lines are used less frequently and maintain high immunogenicity similar to chemically induced models [48]. These are most commonly engineered by injecting lentiviral retroviral vectors into mice that express oncogenes [49].

In comparison, spontaneous tumor models are relatively rare and can be challenging to cultivate [50]. The VM mouse strain is unique, as its inbred properties result in a relatively high incidence (1.5%) of spontaneous brain neoplasms. As these tumors develop more naturally, they mimic the invasive growth characteristics of human GBM. VM-M3 and SMA-560 are two examples of glioma cell lines that were derived from this mouse strain [51,52].

As previously mentioned, transposons may be leveraged to genetically engineer various cell types. The SB transposon system is part of the Tc1/mariner class and can recognize specific DNA sequences known as inverted repeat/direct repeat (IR/DR) sites. After recognition, they are able to ‘cut and paste’ to integrate DNA transposons into the host genome at these sites. This method has been employed to generate novel glioma models in mice by inserting genetic lesions into the genome of stem cells along the lateral ventricle of neonatal mice [53,54,55,56]. By altering the murine genome, the etiology and histopathology of human GBM are closely recapitulated [57]. These tumor cell lines can be engineered to express tumor-specific antigens such as EGFRvIII, IL13Rα2 and GD2 to create CAR T-cell glioma models. Below, we discuss the most relevant murine glioma cell lines to be used for CAR T-cell therapy research.

##### GL261

The GL261 cell line originated in 1970 through chemical induction, when methylcholanthrene pellets were surgically implanted into the brains of C57BL/6 mice [58]. Despite its widespread use, a disadvantage of GL261 is its robust immunogenicity compared to human GBM [59,60]. GL261 cells have elevated MHC I expression and tumor mutational burden (TMB). The TMB from GL261 is almost 5000 mutations per Mb, while human GBM has an average of 2.7 mutations per Mb [61]. High TMB has been shown to correlate with better responses in models of immune checkpoint blockade, which may, in part, be responsible for the disparity between pre-clinical promise and success in clinical trials for these agents [62,63]. Furthermore, GL261 cells contain mutations of p53 and K-Ras, leading to elevated levels of c-Myc, which is not typically observed in human GBM [59].

##### CT-2A

CT-2A was created in 1992 by chemical induction using 20-methylcholanthrene [64]. It is believed to be more aggressive than other established glioma cell lines [65]. Similar to GL261, CT-2A is used extensively in GBM research, given its ability to mimic intra-tumoral heterogeneity, as well as the radio- and chemoresistance that is associated with GBM [66].

##### SMA-560

SMA-560 was not generated with chemical induction but was, instead, derived from spontaneous tumors in 1980 [52,67]. SMA-560 is moderately immunogenic due to modest expression of MHC I [68]. The murine background of this cell line is VM/Dk. SMA-560 is moderately used—but far less frequently than GL261 and CT2A—and is, therefore, also, perhaps, less well characterized [69].

##### SB28

SB28 was generated in 2014 using sleeping beauty (SB) transposon constructs that induced de novo gliomas in neonatal C57BL/6 mice [54,70]. SB28 has been shown to be low in immunogenicity and is negative for MHC I [71]. Furthermore, compared to all other known murine GBM cell lines, SB28 displays one of the most accurate TMEs relative to human GBM. SB28 is almost fully resistant to treatment with immune checkpoint blockade and has a lower mutational load [62].

##### 005

The 005 murine GBM cell line was derived from glioma stem cells after lentiviral transduction with H-Ras and AKT in Trp53+/− C57BL/6 mice in 2009 [72]. Relative advantages of this model include its reduced immunogenicity and histological similarities to human GBM with regard to heterogeneity and invasiveness. Recently, 005 tumor models, when compared to GL261, CT-2A and Mut-3, were found to most closely resemble the immune-phenotypic environment of human GBM specimens [73].

##### 4C8

In 1999, another new murine glioma cell line was developed utilizing glioma cells from a spontaneous glioma grown in a transgenic B6D2F1 mouse [74]. 4C8 has proven to be a highly cellular tumor with aggressive invasion into cerebral ventricles and meninges (Table 1).

#### 3.1.2. Human Cell Lines for GBM

Human cell lines for GBM can be used both in vitro and in vivo in xenograft models. Xenograft models of GBM involve the engraftment of human GBM cells into immunodeficient mice. The following three types of immunodeficient are primarily used: (1) nude mice, which lack T cells; (2) non-obese diabetic severe combined immunodeficiency (NOD-SCID) mice, which lack T and B cells; and (3) NOD-SCID IL2R-γ null (NSG or NOG) mice, which lack T, B or NK cell activity [75,76]. Disadvantages of xenograft models include the inability to study the effects and impact of the endogenous immune system.

By engrafting patient-derived GBM cells into immunocompromised mice, the histopathologic, genomic and phenotypic characteristics of the primary tumor are preserved. The wide variety of human gliomas that have been tested in these models have been catalogued elsewhere [77,78,79]. In general, xenograft models offer the advantage of using human cancer cells, providing more accurate insights into aspects of tumor development and therapeutic response to therapies that can be used in patients. However, these models lack an endogenous immune system.

## 4. Overview of In Vitro Models

### 4.1. Cell Culture Assays

CAR T-cell efficacy is often correlated with the ability of a particular approach to induce T-cell activation, persistence, proliferation, inhibition and exhaustion. Perhaps the most straightforward method of evaluating CAR T cells in an in vitro system is a cytotoxicity assay. In these assays, effector response is measured against cells expressing tumor-associated antigens (TAAs). The killing of target cell lines can be quantified and analyzed using live cell imaging, bioluminescence assays or flow cytometry analysis [80]. The chromium release assay, developed in the 1968, provides an alternative approach [81]. The co-culturing of tumor target cells with CAR T cells at different densities can also be used to assess CAR T-cell proliferation, activation and degranulation. With the addition of certain agents, such as 3H-thymidine or presto blue, fluorescence intensity can be measured after several days as a proxy for proliferation and expansion. T-cell activation, degranulation and differentiation can also be evaluated by flow cytometry. These assays offer efficient, inexpensive and valuable methods of characterizing CAR T-cell therapies.

### 4.2. Patient-Derived Organoids

Cerebral organoid models represent an attempt to better recapitulate the environment of the brain. They are three-dimensional cell cultures derived from a patient’s tumor and used to model the TME, although earlier versions were generated from normal tissue organoids by introducing pro-tumorigenic mutations [82,83]. By utilizing induced pluripotent stem cells or embryonic stem cells and a specialized growth process, a primitive organoid resembling certain structures in the brain can be cultured in vitro [84]. These organoid models retain key features of the patient’s tumor, such as its heterogeneity and microenvironment [85]. They serve as an intermediary between in vitro and in vivo models.

### 4.3. Spheroid Model

Spheroids are 3D cell aggregates that are not attached to any surface, providing a unique environment that more accurately mimics solid tumors compared to 2D cell cultures. Spheroid cells are often cultured with a hydrogel matrix, which functions as an extracellular matrix [86,87]. Multiple spheroid models have been constructed for GBM to study immunological approaches [86,88,89]. Spheroids are either matrix-based or matrix-free (i.e., suspension cells). Matrix-based models provide the advantage of demonstrating the invasion of tumor cells [90]. Generally, these scaffolds are made up of natural or synthetic hydrogels in an effort to mimic the brain ECM.

### 4.4. Organ-on-a-Chip Model

Organ-on-a-chip (OoC) models also seek to incorporate critical features of the human TME [91]. Inside each chip is a co-culture of primary human cells with a tissue–vascular interface. Media flowing in the vascular channel provide shear stress, which is required to model human immune cell recruitment with specificity [92]. For instance, CAR T cells may flow through these channels and migrate through the endothelium in response to local inflammatory conditions. Limitations of OoC models include the fact that the chip materials are artificially engineered and may not exactly replicate the TME. The absence or imbalance of specific cellular components can also lead to significant differences between models [93].

## 5. Overview of In Vivo Models

Prior to translation to clinical trials, in vivo models are essential in the evaluation of the efficacy and safety of CAR T-cell therapies for brain tumors. To date, the most commonly used models are syngeneic and xenograft murine models. Both have unique advantages and limitations, and no single approach provides a perfect pre-clinical framework. Recent advances and new models such as transgenic or humanized mice, canine models and non-human primate models provide novel ways to evaluate CAR T-cell therapies (Figure 2).

### 5.1. Syngeneic Mouse Model

The syngeneic mouse model is the oldest and most utilized model for immunotherapy [94]. A syngeneic mouse model uses immunocompetent mice in which histocompatible tumor cells are implanted [95]. These tumor cells are derived from a genetically identical murine strain. Orthotopic inoculation provides a more realistic TME [96]. The site of injection can trigger variable antitumor responses [97].

The advantages of a syngeneic model includes a functional immune system; and TME and, in certain cases, the ability to study pre-conditioning therapy prior to adoptive transfer [69,98]. This model is also particularly well-suited for combination regimens of CAR T-cell therapy and other immunotherapies, such as oncolytic viruses (OV), checkpoint inhibitors or small-molecule inhibitors [99,100]. However, syngeneic models have not been especially useful in other settings, such as in the recapitulation of cytokine release syndrome (CRS), the most commonly found toxicity in patients treated with CAR T-cell therapy. Furthermore, syngeneic tumor cells, similar to all cell lines cultured in vitro, are subject to mutations and genetic drift over time.

### 5.2. Xenograft Mouse Model

Xenografts from cell lines and patient-derived xenografts (PDXs) for GBM have been extensively utilized to recapitulate molecular and genomic characteristics of human tumors and to assess therapeutic efficacy in the setting of challenges such as tumor heterogeneity [101].

#### 5.2.1. Xenografts from Cell Lines

Cell line-derived xenografts, particularly from the U87 and U251 glioblastoma cell lines, have been a mainstay in brain tumor research for decades [102]. These models involve the transplantation of human tumor cell lines into immunocompromised mice, either in the brain (orthotopic) or in other tissues (heterotopic), achieved by subcutaneously injecting cells into the flank, for example. Typically, severe combined immunodeficient (SCID) [103], non-obese diabetic (NOD) [104] and NOD-*SCID IL2rγ^null^* (NSG) mice are used to establish xenografts [105,106]. As a caveat to the use of these models, one study found that the U87 cell line had a drastically different DNA profile from that of the original tumor it was generated from [107]. Additionally, many of these cell lines have demonstrated failure to infiltrate the brain parenchyma when orthotopically implanted [108].

#### 5.2.2. Patient-Derived Xenografts (PDXs)

Patient-derived xenograft models represent a significant advancement over cell line-derived xenografts, as they may maintain the cellular heterogeneity and architecture of the original patient tumors. PDX models are established by transplanting fresh tumor tissue or cells obtained from patient tissue directly into immunocompromised mice [101].

While cells can also be grown in adherent culture, and studies have shown superior tumor engraftment of neurospheres injected into the brain [109,110]. PDXs generated by these methods have been used in GBM CAR T-cell experiments [28,111]. One PDX, namely BT74, was used to accurately recapitulate glioma heterogeneity and the physiological expression of two GBM antigens, namely EGFRv3 and IL13Rα2, in pre-clinical studies of CAR T-cell approaches [111]. Despite these translational advantages, PDXs are laborious to generate and maintain, and engraftment rates for GBM range from 25 to 75 percent [78,102,112,113,114].

### 5.3. Transgenic Mouse Model

Transgenic (Tg) mouse models or genetically engineered mouse models (GEMMs) are unique models for the study of the effects of CAR T-cell therapy. In these models, the phenotype of a mouse can be engineered, for example, by stimulating the expression of oncogenes or inactivating tumor-suppressor genes [115]. Transgenic models insert oncogenes into tumor precursor cells to obtain carcinogenesis in mice, and genetically engineered mouse models broadly refer to these and other strategies used for genetic perturbation [116]. Genetically engineered mouse models can be obtained through either viral delivery, the Cre-LoxP system, the RCAS-TVA system, CRISPR-Cas9 or transposons [88]. This means tumor growth in the murine model occurs more naturally than that of artificially implanted cancer cells. As these tumors would grow in situ and are not implanted, which provides a unique environment to study tumor progression [117]. Additionally, specific tumor mutations and TAAs can be incorporated into the tumor generation process to allow for more specific murine models [118]. Numerous GEMM glioma cell lines have been created [116,119]. Key published GEMMs are summarized in Table 2.

### 5.4. Humanized Mouse Model

A humanized mouse has no murine immune system but is engrafted with a human immune system. The most commonly used model is NSG (NOD SCID) mice transplanted with human CD34+ HSPCs or PBMCs [131,132]. HSPCs can regenerate lymphoid and myeloid compartments, but T-cell development in these animals is not optimal due to the lack of a thymus [133]. It is also possible to inject human CD34+ cells without thymectomizing the mice [134]. Since these mice lack T, B, NK and functional DCs, they are highly receptive to engraftment of human cells—both healthy and tumor tissues [135]. Another approach to this model requires the implantation of bone marrow, liver and thymus (BLT) into immunodeficient mice, typically NSG. This favors the lymphoid compartment and results in improved T-cell reconstitution, maturation and selection [136].

### 5.5. Canine Model

One infrequently used approach for CAR T-cell therapy for gliomas is the use of canine models, which typically arise in genetically outbred animals with intact immune systems [137]. Canine tumors are quite similar to human tumors and have shown comparable responses to various treatments [138]. Many of the adversities seen in clinical trials have also been observed in canine trials, reinforcing the similarity between humans and dogs [139]. However, canine trials require unique expertise, funding and specific infrastructure.

### 5.6. Primate Model

A few groups have recently tested CAR T-cell therapies using non-human primate (NHP) models. One relative strength of NHP models is their ability to provide greater fidelity when recapitulating immune system characteristics and genetic heterogeneity. These similarities give primate models an advantage when testing the safety of CAR T-cell therapies, specifically for on-target/off-tumor toxicity and cytokine release syndrome (CRS) [133].

A limitation of NHP systems is the dearth of tumor models available for meaningful therapeutic evaluation. Primary tumors in NHP models are rare and have limited practicality. The use of tumor cell lines from other animals would also quickly lead to rejection of the tumor [140]. As such, NHP models may be better suited for the modeling of the toxicity and safety of CAR T-cell therapies.

The first study evaluating CAR T cells in NHP models was conducted in 2015 [141]. Researchers tested CAR T cells targeting orphan tyrosine kinase receptor ROR1, which is frequently expressed in many hematological and solid malignancies. They observed no symptoms of toxicity after treating two macaques with autologous CAR T cells, even at high dosages [142]. As of April 2024, there are five ongoing clinical trials investigating the safety of ROR1-targeting CAR T cells in humans [143]. Another group performed a study on non-human primates using an anti-CD20 CAR T cell therapy. They found signs of significant cytotoxicity correlated with pro-inflammatory CSF cytokines and pan-T-cell encephalitis. Evidence of CRS included levels of IL-6 and IL-8 that were recapitulated in clinical trials [144]. Ephrin type B receptor 4 (EPHB4) CAR T cells were also tested in a lymphodepleted NHP model. Researchers found a slight elevation in IL-6 but no symptoms of CRS. This might be attributed to this certain model having no tumor and, therefore, no TAA to fully activate the CAR T cells. This could prevent CRS or immune effector cell-associated neurotoxicity syndrome (ICANS) [145]. To date, no CAR T-cell therapies targeting brain tumors in NHP models have been described, but previous studies show their promising utility in modeling the safety of these therapies.

## 6. Recommendations

The authors recommend considering both xenograft and syngeneic in vivo murine models for the pre-clinical validation of CAR T-cell therapies for glioma in order to enhance the evaluation of both specificity and toxicity. Clinical trials with CAR T-cell therapies for solid tumors have highlighted the importance of the TME, underscoring the value of models that can recapitulate the TME. There should be some consideration given to the planning of experiments that match in vitro and in vivo systems. These models will become essential in the future for the establishment of more reliable CAR T-cell therapies. We recommend that researchers critically evaluate all models and consider the specific goals of investigation of their therapies.

## 7. Conclusions

The integration of different pre-clinical models is essential for the advancement of CAR T-cell therapy for brain tumors towards clinical success. While each model contributes valuable insights, none is able to fully replicate the complexities of human brain tumors. Researchers must carefully consider the benefits and limitations of each model for pre-clinical evaluation of their therapies. Although imperfect, in vitro and in vivo models have proven vital in the discovery and development of new treatment opportunities for brain tumors. Future efforts should focus on refining existing models, developing novel systems and, most, importantly maintaining collaboration between pre-clinical and clinical programs.

## Figures and Tables

**Figure 1 cells-13-01480-f001:**
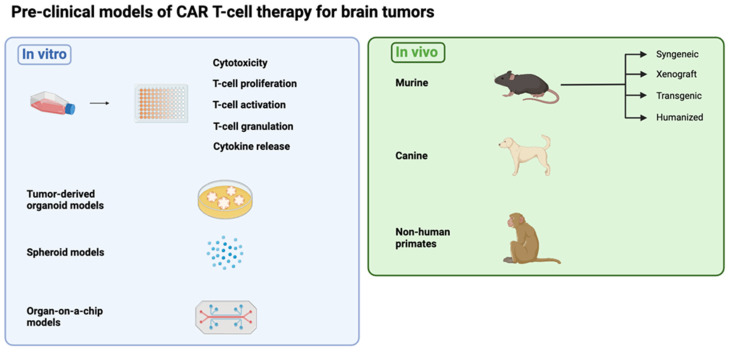
Overview of the pre-clinical models discussed in this review. (Figure created with BioRender.com (accessed on 3 April 2024)).

**Figure 2 cells-13-01480-f002:**
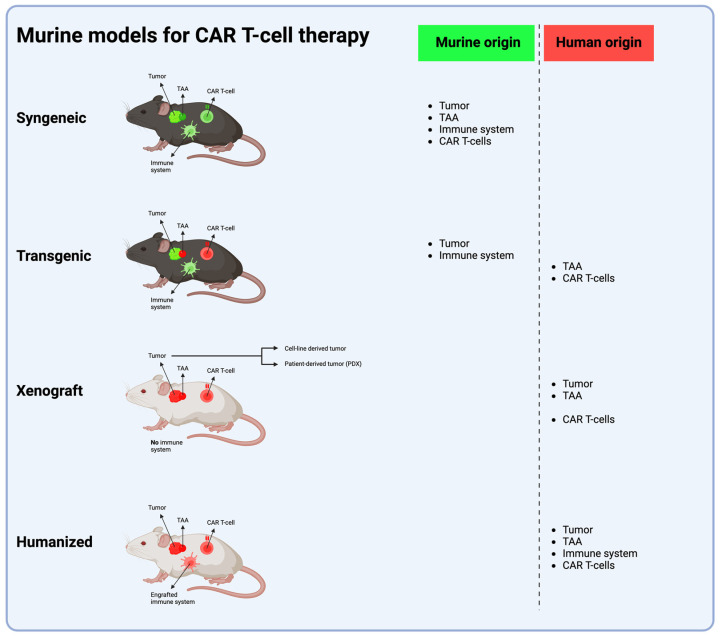
The four murine models most frequently used in CAR T-cell therapy development. The syngeneic model uses an immunocompetent mouse with a functional murine immune system implanted with a tumor of murine origin; this tumor holds a murine TAA, which is targeted with murine CAR T cells. The transgenic model utilizes an immunocompetent mouse genetically engineered to express a human TAA, which is targeted with human CAR T cells. Xenograft and humanized models both use immunoincompetent mice, with NSG mice being the most widely used. In the xenograft model, in vitro cultured human tumor cells or patient-derived human cells are implanted into mice lacking a functional immune system, and the mice are, thereafter, treated with human CAR T cells. In the humanized model, (a part of) the functional human immune system is engrafted into the mouse, human tumor cells are implanted and the mouse is treated with human CAR T cells. (Abbreviations: TAA = tumor-associated antigen) (Figure created with BioRender.com (accessed on 1 April 2024)).

**Table 1 cells-13-01480-t001:** Syngeneic glioma cell lines.

Cell Line	Method of Generation	Murine Strain	Benefits	Limitations
GL261	Chemical induction	C57BL/6	Highly proliferative Well characterized	High TMB Vulnerable to genetic drift Highly immunogenic Lacks MMR genes → TMZ resistance
CT-2A	Chemical induction	C57BL/6	Highly proliferative Well characterized	High TMB Vulnerable to genetic drift Modest invasion in brain parenchyma
SMA-560	Spontaneous tumor	VM/Dk	Spontaneous generation	Less well characterized VM/Dk background Immunogenic
SB28	Sleeping Beauty (SB) transposons	C57BL/6	Most accurate TME of all cell lines Poorly immunogenic	Less well characterized
005	Lentiviral induction	C57BL/6	Poorly immunogenic Accurate TME	Less well characterized
4C8	Spontaneous tumor	B6D2F1	Spontaneous Robust invasion in surrounding brain tissue	B6D2F1 background

**Table 2 cells-13-01480-t002:** GEMMs of glioma cell lines.

Genes	Method of Generation	Tumor Type	Reference
*NF1*, *PTEN* and *Trp53* loss	CRISPR-cas9	GBM	[120]
*H3.3^K27M^* and *Trp53* loss	CRISPR-cas9 and transposons	GBM	[121]
*PDGFB* expression; *Trp53*, *PTEN* or *CDKN2a* loss	CRISPR-cas9	GBM	[122]
*EGFR* activation and *CDKN2a*	RCAS-TVA	Low-grade glioma	[123]
*Kras^G12D^* and *Akt*	RCAS-TVA	GBM	[124]
*PDGFB* expression; *Chk2*, *ATM* or *Trp53* loss	RACS-TVA	GBM and low-grade glioma	[125]
*Trp53* and *PTEN* loss	hGFAP-Cre	Grade III and IV glioma	[126]
*EGFRvIII*, *Ink4a* and *PTEN*	Cre	GBM	[127]
*Idh1^R132H^*	Cre	Glioma precursor	[128]
*Trp53* and *PTEN* loss	Adenoviral-cre	GBM	[129]
*Trp53*/*NF1* loss	SynI-cre and CamK2a-cre	GBM	[130]
*Atrx*, *Trp53* loss and *Nras* overexpression	SB transposon	GBM	[56]

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
