# Peer review of "Pre-Clinical Models for CAR T-Cell Therapy for Glioma"

_cells, 2024, doi:10.3390/cells13171480_

Round 1

Reviewer 1 Report

Comments and Suggestions for Authors

The authors outline the various in vitro and in vivo preclinical model systems that can be used for the investigating CAR T-cell therapies. Overall, the paper is comprehensive although not particularly detailed if the purpose of the manuscript is to help researchers evaluate and select the best model systems for investigating CAR T-cell therapies.

Three specific areas where I would like to see a more detailed discussion include:

1) the role for more sophisticated transgenic mouse models that allow for spontaneous tumor formation that incorporates specific tumor antigens/mutations. An excellent reference for this technology is:

Kim GB, Rincon Fernandez Pacheco D, Saxon D, Yang A, Sabet S, Dutra-Clarke M, Levy R, Watkins A, Park H, Abbasi Akhtar A, Linesch PW, Kobritz N, Chandra SS, Grausam K, Ayala-Sarmiento A, Molina J, Sedivakova K, Hoang K, Tsyporin J, Gareau DS, Filbin MG, Bannykh S, Santiskulvong C, Wang Y, Tang J, Suva ML, Chen B, Danielpour M, Breunig JJ. Rapid Generation of Somatic Mouse Mosaics with Locus-Specific, Stably Integrated Transgenic Elements. Cell. 2019 Sep 19;179(1):251-267.e24. doi: 10.1016/j.cell.2019.08.013. PMID: 31539496; PMCID: PMC6934691. 

2) How to generate the CAR T-cells themselves that can be used for studies with mouse syngeneic cell lines given the need for a murine CAR antigen target. An excellent reference for one approach to overcome this is: https://pubmed.ncbi.nlm.nih.gov/34347086/

3) A table summarizing CRS/ICANS rates seen in the currently published CAR T-cell preclinical literature given the critical importance these adverse events will play in the clinical translation of these therapies. 

Reviewer 2 Report

Comments and Suggestions for Authors

This review overviews different mouse models for gliomas. The study describes the types of available pre-clinical models for CAR T cell studies in gliomas. This is a timely and relevant and practical review that will be useful for CAR T labs to select appropriate models.  I have a few comments that could be considered to strengthen the work:

Concerns

1.     After 5th paragraph of the abstract, the footnote is placed between the sentence, should it be placed after punctuation?

2.     Second background paragraph seems redundant to the information regarding the barriers of solid tumors, already spoken about in the first paragraph

3.     For the murine glioma models section, maybe adding a sentence for the transition or an introduction into the description of each cell line would be good.

4.     Should success of TILs in melanoma be mentioned and considered?

5.     Line 67 – Globlastoma “multiforme” is an outdated term.  It should just be Glioblastoma.

6.     Line 68 - WHO grade “4” instead of WHO grade “IV” is the new preferred nomenclature in the most recent WHO system

7.     Lines 68 – the cell of origin for GBM may be a neural stem cell, radial glia, or progenitor – this may be controversial/incompletely understood – it may be better to state that the GBMs are thought to arise “from glia or related progenitor cells.”

8.     The is some flip-flopping in terminology between GBM and glioma which might be able to be streamlined for clarity/focus.  It may be best to put in a few sentences in the intro defining these overlapping entities: GBM vs. glioma vs. malignant glioma vs. DIPG etc.

9.     97 – analog should be “analogous”

10.  Section 2.1 is titled “cell lines” but it really focuses on “cell lines used for syngeneic or xenograft models” – perhaps the title can be made more accurate.

11.  The work may benefit from a figure (or two) summarizing the different in vitro assays and in vitro models described in section 3

12.  The authors may choose to synthesize their review with a “Recommendations” paragraph describing which models that the PI’s lab chooses to use and some concrete recommendations for labs pursuing the research.

13.  The description of GEMMs and Transgenic lines is rather short and vague.  There are numerous such lines.  Some of them could be described, and a table could be provided to overview the different lines.  At a minimum perhaps a good review of the exisiting glioma GEMM lines could be provided.

14.  There is a slight confusion about nomenclature of GEMMs vs. Transgenics.  Are they the same thing for the purpose of this review?  The figure implies that “transgenic” specifically applies to models in which the tumor antigen is introduced transgenically – does that mean GEMMs are the broader group of models that may use genetic perturbations to drive tumorigenesis, but don’t specifically express a tumor antigen using the genetic perturbation?

15.  Should some consideration be given to planning experiments in terms of matching in vitro models with in vivo models?

16.  The authors could mention TIL therapy, recently approved for select melanoma patients, and discuss how these considerations and models may apply to TIL therapy experiments (presumably pretty similar but there might be additional considerations?).

-Zach Reitman

Comments on the Quality of English Language

Mostly solid.

Reviewer 3 Report

Comments and Suggestions for Authors

The review by Vandecandelarere et al. attempts to acquaint the novice reader about different models available for glioma research.

It’s written for people not familiar with the field, and doesn’t go into much depth.

From the title “Pre-clinical models for CAR T cell therapy for glioma” I expected much more information on how to actually set up and test new candidate  CAR T cell based therapeutics, or new drugs to enhance CAR T cell therapy, for gliomas.

For instance, if I wanted to test new therapeutics for glioblastoma (GBM) in a mouse model, which GBM cell lines and CAR T cells should I consider to use as a model system.? Yes they mention GL261, CT2A etc….But, which CAR T cells do I pair them with? Where and how do I obtain the different CAR T cells? Are there transgenic mouse lines available from Jax that will provide a source of CAR T cells for a specific murine GBM cell line? Do I need to use retroviral transductions?

Please greatly expand on advising the scientific community on how to practically perform research with GBM-reactive CAR T cells in mice.

Round 2

Reviewer 2 Report

Comments and Suggestions for Authors

All concerns addressed.  

Comments on the Quality of English Language

Editors and authors should check for minor typos (RCAS not RACS in the table, etc).

Reviewer 3 Report

Comments and Suggestions for Authors

I thank the authors for their efforts towards addressing the question of how to actually set up CAR studies…which was also the main suggestion of Reviewer #3. However, their additions are in very broad-brush strokes (only a few sentences for each technique) leaving the reader pretty uninformed about how to actually set up and perform CAR T cell studies. What other CAR T transgenic mice besides those in reference  #38 might be useful? Which tumor cell lines (hopefully obtainable from a commercially available source) and readily available CAR T cells can be paired with so that researchers new to the field can test their candidate adjunct therapeutics for mouse GBM?

The authors give the strong impression that these studies are very complicated, and a great deal of effort and expertise are necessary to pursue such studies. So, Im left thinking that the only way to pursue such studies is to collaborate with an expert, which is contradictory to the main purpose of this this review, which is to inform the scientific community about CAR T cell models so that other researchers can do these studies themselves. If the authors can give more detailed and instructive advice, it will be appreciated. Although they provide a few references, these examples are complex and again make the reader only want to seek collaboration with experts, which very few people can do.

Since GBM models have already been reviewed numerous times, I think providing additional guidance in the manuscript would be an important new contribution to the field, will assist newcomers to the field, and lead to their publication being more widely read and cited.
